# Regulation of *INSM1* Gene Expression and Neuroendocrine Differentiation in High-Risk Neuroblastoma

**DOI:** 10.3390/biology15010022

**Published:** 2025-12-22

**Authors:** Chiachen Chen, Siyuan Cheng, Xiuping Yu, Yisheng Lee, Michael S. Lan

**Affiliations:** 1Department of Genetics, School of Medicine, Louisiana State University Health Sciences Center, New Orleans, LA 70112, USA; 2Department of Biochemistry & Molecular Biology, LSU Health Shreveport, Shreveport, LA 71103, USA; 3Foresee Pharmaceuticals, Newark, DE 19713, USA

**Keywords:** neuroblastoma, INSM1, N-Myc, sympatho-adrenal lineage, neuroendocrine differentiation, methionine cycle signaling, SAM, SAH, 5′-iodotubercidin

## Abstract

Neuroblastoma (NB) is a childhood cancer that arises when certain nerve-related cells fail to mature properly and become malignant. Our study focuses on a gene called INSM1, which is normally active during early cell development but is abnormally elevated in NB tumors. We aimed to understand how INSM1 contributes to cancer’s aggressiveness and poor differentiation. We discovered that INSM1 keeps tumor cells in an immature, stem-like state and is closely linked to a chemical pathway called the methionine cycle, which affects gene regulation through epigenetic changes. Disrupting this cycle alters INSM1 activity and may help push cancer cells toward normal development. We also found that retinoic acid, a compound known to promote cell differentiation, reduces INSM1 and another cancer-driving gene, N-Myc. These findings suggest that targeting INSM1 and its metabolic regulation could be a promising strategy to treat NB by encouraging tumor cells to mature and lose their malignant properties. This research provides new insights into the biology of NB and may lead to more effective therapies for children affected by this disease.

## 1. Introduction

Neuroblastoma (NB) is an aggressive pediatric malignancy originating from granule neuron precursors of the sympatho-adrenal (SA) lineage [1,2]. Metastatic NB occurs in ~70% of cases, with high-risk patients facing a survival rate of ~40% and a long-term survival rate close to 15% [3]. A major contributor to this poor prognosis is intra-tumoral heterogeneity, the diverse subpopulations of cancer cells that differ in morphology, gene expression, metabolism, proliferation, motility, and metastatic potential. In NB, cellular heterogeneity is tightly linked to variable neuroendocrine differentiation (NED) states, which sustain malignant plasticity and resistance to therapy [4]. N-Myc is a well-established driver of aggressive NB, yet direct targeting N-Myc remains challenging due to its intrinsically disordered structure and essential roles in normal cellular processes [5,6,7]. As a result, there is a critical need to identify novel nodes that cooperate with N-Myc to sustain NB progression. Our previous work identified INSM1 (insulinoma-associated protein 1) as a critical downstream effector of N-Myc. We have shown that N-Myc transcriptionally activates INSM1, and in turn, INSM1 stabilizes N-Myc protein, forming a feed-forward loop that promotes tumor growth and aggressive malignancy [8]. INSM1 expression promotes tumor growth, NED, and aggressive malignancy, while its inhibition reduces tumorigenicity in vitro and in vivo [9,10]. Additionally, we identified an adenosine kinase (ADK) inhibitor, 5′-iodotubercidin (5′-IT) potentially inhibits INSM1 expression and NB tumor growth [11]. These findings position INSM1 as a central node in N-Myc driven oncogenic program and support further investigation into its regulatory functions and NED of high-risk NB [12].

NB presents a heterogeneous pediatric malignancy characterized by variable differentiation and clinical outcomes, with poorly differentiated tumors often associated with high-risk disease and aggressive behavior. Recent evidence highlights the transcription factor INSM1 as a critical regulator in NE development and tumor biology [9]. Our preliminary data demonstrate that INSM1 expression is markedly elevated in poorly differentiated, high-risk NB tumors compared to more differentiated counterparts [12]. Functionally, INSM1 appears to sustain the NED state, promote tumor cell proliferation, and enhance malignant potential, thereby contributing to aggressive disease phenotypes [8]. These findings suggest that INSM1 not only serves as a biomarker for poor prognosis but may also represent a promising therapeutic target for intervention in aggressive NB subtypes. Together, these studies established that INSM1 is a critical regulator of NB malignancy and a potential vulnerability for intervention.

In the current study, 5′-IT suppresses *INSM1* promoter activity, shifts intra-/extra-cellular adenosine levels, induces adenosine receptor A3 (ARA_3_) signaling pathway, and triggers an apoptotic state in NB tumor cells [11]. However, upon 5′-IT treatment of NB cells there is a gap in knowledge between adenosine imbalance and *INSM1* promoter suppression. Gene expression patterns usually show silencing in heterochromatin and activation of expression in euchromatin as DNA methylation or histone modification plays a key role [13]. Interestingly, we uncovered a MAT2α inhibitor (FIDAS-5) that blocks SAM production and also enhances *INSM1* promoter activity, in contrast to the negative effect seen with 5′-IT [14]. These two-end of methionine pathway inhibitors (5′-IT and FIDAS-5) implicate the SAM to SAH conversion with the opposite effects on *INSM1* promoter activity. The SAM to SAH conversion suggests a critical role via the modulation of the trans-methylation reaction surrounding the *INSM1* promoter locus. Normally, SAHH hydrolyzes SAH into adenosine and homocysteine. This prevents SAH accumulation, which otherwise inhibits methyltransferases. When SAHH is inhibited by D9 (an analog of DZNep), SAH levels rise [15]. SAH is a potent feedback inhibitor of SAM-dependent methyltransferases, reducing methylation reactions across DNA, RNA, and proteins [16]. This leads to global hypomethylation, affecting gene expression and epigenetic states.

To dissect the molecular mechanism underlying *INSM1* gene regulation, the effect of 5′-IT to induce adenosine imbalance was examined in relationship with its perturbation of methionine cycle flux. The methylation index (SAM/SAH) change should either suppress or activate *INSM1* promoter activity via epigenetic modifications. While INSM1 has been recognized as a marker in NE tumors, its functional role in NB progression remains largely unexplored. Our study is the first to demonstrate that INSM1 is not merely a diagnostic marker but an active driver of aggressive phenotypes at high-risk, poorly differentiated NB. By linking INSM1 expressions to maintenance of the NED state and enhanced tumor proliferation, we uncovered a previously uncharacterized mechanism contributing to NB malignancy. This positions INSM1 as both a prognostic biomarker and a potential therapeutic target, offering new avenues for precision treatment in aggressive NB subtypes.

## 2. Materials and Methods

### 2.1. Cell Culture and Reagents

Seven human NB cell lines were selected based upon the expression patterns of INSM1 and N-Myc. Human NB cell lines, SKnSH, SH-SY-5Y, CHP212, SKnBE-2, CHP134, IMR-32, BE2-M17 were obtained from American Type Culture Collection (Manassas, VA, USA). NB cells were cultured in RPMI-1640 medium supplemented with 10% fetal bovine serum (Atlanta Biological Inc., Flowery Branch, GA, USA), 1× penicillin/streptomycin in 5% CO_2_ incubator at 37 °C. All the cell lines were authenticated with Short Tandem Repeat DNA profiling analysis. The passage number of the cells used for the experiments was kept under 30. 5′-iodotubercidin (5′-IT, HY15424), 3-Deazaneplanocin (D9, an analog of DZNep-HCl, HY10442), and FIDAS-5 (HY-136144) were obtained from MedChemExpress (Monmouth Junction, NJ, USA). Antibodies, INSM1 (A8, sc-271408), N-Myc (B8.4B, sc-53993), were purchased from Santa Cruz Biotechnology (Dallas, TX, USA). LSD1 (C69G12) and EZH2 (D2C9) antibodies were purchased from Cell Signaling Biotechnology (Danvers, MA, USA). The antibody to tubulin and β-tubulin III were purchased from Life Technologies (Chicago, IL, USA) and the β-actin antibody was obtained from Sigma (St. Louis, MO, USA). Horseradish peroxidase-conjugated secondary antibody was obtained from Bio-Rad laboratories (Hercules, CA, USA).

### 2.2. Analysis of SAM/SAH by ELISA

BE2-M17 cells (1 × 10^7^) were treated with 5′-IT (1.0 µM), DZNep (10 µM), or FIDAS-5 (1 μM) for 30 min or the indicated time point. Collect cells by centrifuging at 2000× *g* for 10 min at 4 °C. Sonicate or homogenize the cell pellet on ice in 1–2 mL cold PBS. Centrifuge at 10,000× *g* for 15 min at 4 °C. Remove the supernatant and store it on ice. Aliquot and store the supernatant at −80 °C for used in the assay. Standard SAM (0–40 nM), SAH (0–12.5 µM), and extracted samples were subjected to SAM and SAH ELISA kit (Cell Biolabs, Inc., San Diego, CA, USA, CBIO-MET-5151-C) assay.

### 2.3. INSM1 Promoter-Driven Luciferase Reporter Assay

Two stable BE2-M17 and IMR-32 cell lines containing an *INSM1* promoter-driven *luciferase* reporter gene were established [11]. Briefly, an *INSM1*-promoter (−426/+40 bp; accession no. U07172) was inserted into pGL4.18-[luc2P/Neo] basic vector (Promega Co. Madison, WI, USA) in front of a *luciferase* (*Luc2*) gene. *INSM1*-promoter reporter plasmid was linearized and transfected into BE2-M17 or IMR-32 cells with Lipofectamine 2000 (Invitrogen, Chicago, IL, USA) for 48 h. Transfected cells were selected with a pre-determined G418 concentration in culture medium for 2 weeks. The G418-selected stable cells were further assayed for the *INSM1* promoter-driven *Luc2* activity. The strong *Luc2* activity reflects strong *INSM1*-promoter activity in each cell line. The *Luc2* activity was analyzed with Firefly Luciferase Assay Kit 2.0 (Biotium Inc., Fremont, CA, USA).

### 2.4. Western Blot Analysis

Cell lysates were extracted with lysis buffer (10 mM Tris-HCl, pH 7.5, 150 mM NaCl, 10% glycerol, 1% Triton X-100, 1 mM DTT, 0.2 mM PMSF, 1 µg/mL aprotinin, 1 µg/mL leupeptin, 1 mM Na_3_VO_4_ and 5 mM NaF) and separated by SDS-PAGE. The electrophoresed proteins were electro-transferred onto a nitrocellulose membrane (Bio-Rad Laboratories Inc.) for Western blot analyses. The membrane was blocked with 5% non-fat dry milk in TBST (20 mM Tris-HCl, pH 7.6, 137 mM NaCl and 0.1% Tween-20), probed with indicated primary antibody overnight at 4 °C, and bound with HRP-conjugated secondary antibody at room temperature for 1 h. The blot was developed with chemiluminescence substrate (Bio-Rad Laboratories Inc.) and exposed on X-ray film (Fuji Film from Z & Z Medical, Cedar Falls, Iowa, USA). The same blot was stripped several times for subsequent antibody blotting. Western blot analysis was repeated in three separate experiments to ensure reproducibility.

### 2.5. Cell Proliferation Assay

BE2-M17 cell proliferation was measured after treatment with 5′-IT (1.0 µM), DZNep (10 µM), or FIDAS-5 (1.0 µM) for five days. Cell counts were determined on each day using trypan blue dye exclusion assay.

### 2.6. RA Induces Neurite-Outgrowth and NB Cell Differentiation

Retinoic acid (RA) induces neuronal differentiation in NB cell lines. IMR-32 and SH-SY-5Y cells were cultured in RPMI-1640 medium supplemented with 0.1% bovine serum albumin (BSA) and treated with 1 µM RA for 5 days. Morphological changes revealed prominent neurite outgrowth in RA-treated NB cells, indicative of neuronal differentiation. Molecular analyses of RA-treated BE2-M17 and SK-N-BE-2 cells were shown in mRNA levels of INSM1 using RT-PCR as compared to untreated controls. Western blot analysis of INSM1 and N-Myc protein levels in IMR-32 cells treated with increasing concentrations of RA (0, 2.5, 5, and 10 µM). β-tubulin III was included as neuronal differentiation marker.

### 2.7. ChIP-Seq Dataset Analysis

Data analysis (GSE162057 and GSE189174) was used for our established pipeline, employing Bowtie2 for sequence alignment, MACS2 for peak calling (narrow peaks for transcription factors and broad peaks for histone modifications), Homer for annotation and motif analysis, and deep Tools for data visualization. To identify genome-wide *INSM1*-binding sites and associated chromatin modifications, we analyze NB cell lines with high endogenous INSM1 expression (IMR-32, Kelly, SK-N-FI, and SKNDZ) and compared them to INSM1-negative lines (GI-ME-N and SH-EP). Chromatin immunoprecipitation followed by sequencing *(ChIP-seq*) datasets were analyzed to map INSM1 occupancy and its interplay with key epigenetic regulators, including EZH2, histone marks H3K27ac, H3K27me3, H3K4me3, and the oncogene *MYCN*.

### 2.8. Transcriptomic RANseq Analysis of INSM1 Overexpression in SH-SY-5Y

SH-SY-5Y cells were infected with adenovirus, *Ad-INSM1* for 48 h to induce overexpression of INSM1. RNA sequencing was conducted using RNA extracted from SH-SY-5Y/*Ad-LacZ* and SH-SY-5Y/*Ad-INSM1* (each containing 2 biological replicates) at Novogene Corporation Inc. (Sacramento, CA, USA). The raw data was first proceeded using FASTP software (Shifu Chen. fastp 1.0) to ensure data quality for further analysis. The raw data was submitted to public domain GEO (accession number GSE312485). The differential expression analysis was performed in R using the DESeq2 package. Gene with an adjust *p*-value less than 0.05 found by DESeq2 were assigned as significantly differential expression. The visualization heatmap was generated in R with heatmap package based on the results from DESeq2 analysis.

## 3. Results

### 3.1. Effects of 5′-IT on INSM1 Expression in NB Cells

Figure 1 illustrates the impact of 5′-IT on INSM1 expression and its relationship with NB cell sensitivity. Panel A shows the chemical structure of 5′-IT, a compound hypothesized to modulate NED pathways. Western blot analysis (Figure 1D, Appendix A) confirmed that among the six NB cell lines tested, CHP212, CHP134, BE2-M17, and IMR-32 exhibited strong expressions of both INSM1 and N-Myc, whereas SK-N-SH and SH-SY-5Y lacked detectable levels of these markers. This differential expression pattern represents diverse heterogeneous genetic background as INSM1 and N-Myc are associated with poorly differentiated, high-risk NB phenotypes. Subsequent treatment of these cell lines with increasing concentrations of 5′-IT (ranging from 5 nM to 20 μM for 48 h) revealed a positive but indirect correlation between cytotoxic sensitivity and baseline INSM1/N-Myc expression (Figure 1B,C). These findings support the hypothesis that INSM1 expression could serve as a biomarker for therapeutic response to 5′-IT and highlight the compound’s potential as a targeted agent against high-risk NB.

### 3.2. 5′-IT Involves in Methionine Cycle Pathway

Methionine adenosyl-transferase II alpha (MAT2α) is a key enzyme that catalyzes the conversion of methionine and ATP into S-adenosylmethionine (SAM), the universal methyl-group donor involved in cellular methylation reactions. S-adenosylhomocysteine (SAH), a byproduct of methylation, acts as a potent inhibitor of transmethylation and is processed through a salvage pathway involving adenosine kinase (ADK) [17]. SAH is reversibly converted to homocysteine by S-adenosylhomocysteine hydrolase (SAHH), allowing for the regeneration of methionine. Homocysteine is then remethylated to methionine using 5-methyltetrahydrofolate (5-CH3-THF) as a methyl donor, a reaction catalyzed by methionine synthase (MTR). Thus, ADK plays a dual role—not only in adenosine metabolism but also as an epigenetic regulator by influencing the efficiency of transmethylation reactions (Figure 2). Methionine pathway inhibitors are 5′-iodotubercidin (5′-IT, ADK inhibitor), DZNep-HCl (SAHH inhibitor), and FIDAS-5 (MAT2α inhibitor).

### 3.3. The Effects of FIDAS-5, DZNep, and 5′-IT on INSM1 Promoter Activities

Our results indicate that the SAHH inhibitor DZNep, which also targets histone methylation, suppresses *INSM1* promoter activity. In contrast, the MAT2α inhibitor FIDAS-5 (F-5), which blocks SAM synthesis, significantly enhances *INSM1* promoter activity. To evaluate these effects, *IMR-32-Luc2* and *BE2-M17-Luc2* NB cells were treated for 24 h with F-5, DZNep, or 5′-IT (an ADK inhibitor), either individually or in combination. As shown in Figure 3A, both DZNep and 5′-IT alone inhibited *INSM1* promoter activity in a dose-dependent manner, while F-5 alone increased promoter activity significantly by at least three-fold. Notably, co-treatment with 5′-IT (0.25 µM) and DZNep (1 µM) produced additive inhibitory effects (*p* > ***) (Figure 3B). However, F-5 was able to partially restore *INSM1* promoter activity significantly when combined with either 5′-IT (Figure 3C) or DZNep (Figure 3D), suggesting that modulation of methionine cycle flux can either suppress or activate *INSM1* promoter activity.

### 3.4. Methionine Cycle Inhibitor on INSM1 Promoter Activity

The regulation of *INSM1*-promoter is closely linked to methionine cycle flux and methylation potential. As shown in Figure 2, MAT2α catalyzes the conversion of methionine and ATP into SAM, the universal methyl donor. After methyl transfer, SAM becomes SAH, which inhibits methyltransferases if it accumulates. Thus, the SAM/SAH ratio is a critical determinant of methylation capacity.

Pharmacological interventions illustrate this relationship. 5′-IT, an ADK inhibitor, disrupts adenosine clearance, leading to SAH accumulation and reduced methylation potential, which suppresses *INSM1* promoter activity. Similarly, DZNep (D9) increases SAH by blocking its conversion to homocysteine, also inhibiting methyltransferases. In contrast, FIDAS-5, a MAT2α inhibitor, lowers SAM synthesis and the SAM/SAH ratio, but paradoxically enhances INSM1 promoter activity (Figure 4E). These effects are specific to the *INSM1* promoter, as *CMVp*-driven *luciferase* remains unaffected (Figure 4A).

SAM/SAH ELISA assays confirm that these compounds disrupt methylation balance in IMR32 cells (Figure 4B–D). Collectively, these findings suggest that methionine cycle perturbations, whether through SAM depletion or SAH accumulation—modulate INSM1 expression via epigenetic mechanisms. Maintaining an optimal SAM/SAH ratio is essential for methylation of DNA, histones, and other substrates, and its disruption alters INSM1 gene regulation and cellular function.

### 3.5. Functional Impact of Methionine Cycle Inhibitors

To evaluate the role of methionine cycle flux in NB cell growth, we performed cell proliferation assays following treatment with three distinct inhibitors: FIDAS-5 (MAT2α inhibitor), 5′-IT (ADK inhibitor), and DZNep (SAHH inhibitor). As shown in Figure 5, these compounds exerted differential effects on NB cell proliferation. FIDAS-5 treatment resulted in a significant increase in cell growth, suggesting that inhibition of MAT2α may relieve a regulatory constraint on INSM1 expression, thereby promoting a proliferative phenotype. In contrast, both 5′-IT and DZNep markedly suppressed cell proliferation consistent with their inhibitory effects on *INSM1* promoter activity observed in earlier experiment (Figure 3A). These findings indicate that methionine cycle enzymes exert opposing influences on NB cell biology, likely through epigenetic mechanisms involving methylation-dependent regulation of INSM1 and associated oncogenic pathways. Collectively, these results underscore the importance of methionine metabolism in maintaining NB aggressiveness and highlight potential therapeutic opportunities through targeted disruption of this pathway.

### 3.6. Proximal INSM1 Promoter (−426/+40 bp) Contains 56 CpG Sites

We analyzed the *INSM1* promoter sequence (−426 to +40 bp) engineered into the *pGL4.18-luciferase* reporter vector that stably transfected into BE2-M17 and IMR-32 cells [18,19]. These *INSM1* promoter-driven *luciferase* cell lines were used to screen small molecules that modulate promoter activity, such as 5′-IT [11]. The *INSM1* promoter region (467 bp) contains 56 *CpG* sites, including an N-Myc binding *E2-box* (*cacgta*) located at −315 to −310 bp (Figure 6). DNA hypermethylation within this *CpG*-rich region may contribute to *INSM1* gene suppression. Notably, a proximal promoter element at −78 to −67 bp (*ctccaggggaag*) serves as an *INSM1* binding site and functions as an autoregulatory suppressor, further influencing *INSM1* gene expression [20].

### 3.7. Epigenetic Modifications of INSM1 Gene Locus

Publicly available *ChIP-seq* datasets from two independent groups (Takenobu and Westermann) were analyzed to examine the epigenetic landscape of the *INSM1* gene locus (chromosome *20p11.23*) across various NB cell lines. *ChIP-seq* analyses rely on public datasets and offer supportive, not definitive, evidence. In IMR-32 cells (INSM1-positive), data from the Takenobu’s group revealed extensive enrichment of multiple epigenetic marks and transcription factors—including EZH2, H3K27ac, H3K27me3, H3K4me3, and MYCN—at the *INSM1* locus (Figure 7A) [21]. Westermann’s group focused on H3K4me3 modifications, comparing INSM1-positive (Kelly, SK-N-FI, SKNDZ) and INSM1-negative (GI-ME-N, SH-EP) NB cell lines [22]. Analysis revealed strong H3K4me3 enrichment at the *INSM1* locus in INSM1-positive NB cells, with minimal modification in INSM1-negative lines (Figure 7B). These findings highlight H3K4me3 as a key epigenetic mark linked to INSM1 expression, suggesting LSD1 involvement through binding to H3K4 di- and tri-methylation (H3K4me2 and H3K4me3) [23].

### 3.8. EZH2 and LSD1 Involvement in INSM1 Gene Expression

To determine whether EZH2 and/or LSD1 contribute to the regulation of *INSM1* gene expression through histone modifications, we examined their expression profiles and potential interactions with the *INSM1* promoter. As shown in Figure 8A Appendix A), treatment with 5′-IT or DZNep significantly reduced *Luc2* reporter activity (0.08/0.45 and 0.02/0.13, respectively), which reflects *INSM1* promoter activity, along with a corresponding decrease in EZH2 expression (0.62 and 0.22). In contrast, FIDAS-5 treatment markedly increased *Luc2* activity (3.07/4.49) while only slightly reducing EZH2 levels (0.94). Notably, both 5′-IT and DZNep treatments led to elevated LSD1 expression (2.64 and 2.52), whereas FIDAS-5 had minimal effect (1.05). These results are consistent with DZNep’s role as a global histone methylation inhibitor, reducing EZH2-mediated H3K27 methylation and thereby suppressing *INSM1* promoter activity [16]. Conversely, increased LSD1—an H3K4 demethylase—may also contribute to repression of *INSM1* transcription. Previous studies have shown that LSD1 interacts with SNAG domain-containing proteins such as INSM1 and GFI1B [24]. To test whether LSD1 is recruited by INSM1 or N-Myc in NB cells, we performed co-immunoprecipitation assays using BE2-M17 cell lysates. Lysates were pre-cleared with agarose-protein G beads and immunoprecipitated with anti-INSM1, anti-N-Myc, or control IgG antibodies. The resulting complexes were analyzed by SDS-PAGE followed by Western blotting with anti-LSD1 antibody (Figure 8B, Appendix A). A 12.5% input lysate served as a positive control, and IgG as a negative control. LSD1 was readily co-precipitated with INSM1, suggesting that INSM1 recruits LSD1 to its promoter binding site (−78/−67 bp, sequence: *ctccaggggaag*). In contrast, N-Myc did not co-precipitate LSD1, indicating specificity in the INSM1–LSD1 interaction.

### 3.9. Role of INSM1 Downregulation in Retinoic Acid-Induced NB Differentiation

To investigate the contribution of INSM1 to the maintenance of poorly differentiated NB, we examined its expression dynamics during retinoic acid (RA)-induced differentiation. RA is a well-established agent that promotes NB cell differentiation and neurite outgrowth [25]. Notably, RA treatment induces neuronal-like differentiation in both INSM1-positive (IMR-32) and INSM1-negative (SH-SY-5Y) NB cell lines (Figure 9A, Appendix A). In INSM1-positive cells, RA stimulation leads to a marked downregulation of INSM1 and N-Myc expression (Figure 9B, Appendix A), suggesting that INSM1 is associated with the undifferentiated, proliferative state of NB cells. Previous studies have shown that knockdown of INSM1 impairs cell invasiveness and reduces tumorigenic potential, further supporting its role in maintaining NB aggressiveness [8]. These findings collectively indicate that INSM1 functions as a critical regulator of NB cell immaturity by maintaining the NED state and supporting proliferative capacity. RA treatment, which induces differentiation of NB cells, significantly downregulates INSM1 expression suggesting that suppression of INSM1 is associated with RA-induced differentiation. Down regulation of INSM1 could be a molecular event in the transition from an undifferentiated, aggressive phenotype to a more mature neuronal state. Mechanistically, this downregulation may involve epigenetic remodeling at the *INSM1* promoter, including changes in histone modifications and transcription factor occupancy, which shift the chromatin landscape toward a differentiation-permissive state. Thus, INSM1 not only serves as a marker of poor differentiation but also represents a functional barrier to RA-mediated maturation, highlighting its potential as a therapeutic target to enhance differentiation-based strategies in high-risk NB. Understanding the transcriptional consequences of INSM1 suppression may provide insight into NB cellular plasticity and uncover novel therapeutic targets aimed at promoting tumor cell differentiation.

### 3.10. RNAseq Transcriptomic Analysis of INSM1 Overexpression in SH-SY-5Y

Retinoic acid (RA) readily induces neuronal differentiation in SH-SY-5Y cells, even in the absence of INSM1 expression. To investigate the transcriptional impact of INSM1, we propose using INSM1-overexpression in SH-SY-5Y cells as an in vitro model for bulk-RNA sequencing to analyze pathway-level changes associated with NB aggressiveness. This approach enables a direct comparison between INSM1-expressing and INSM1-null cells, offering insights into the mechanisms driving NB cellular plasticity and revealing potential therapeutic vulnerabilities. SH-SY-5Y cells were treated with adenoviral INSM1 (*Ad-INSM1*) or control vector (*Ad-LacZ*) for two days, followed by transcriptomic analysis (Figure 10). RA treatment of SH-SY-5Y alone induced neuronal-like differentiation, confirming that INSM1 is not required for this process. However, *Ad-INSM1* overexpression significantly altered the transcriptome compared to *Ad-LacZ* controls. INSM1 overexpression was associated with upregulation of NE and thyroid hormone-related genes (*CHGA*, *CHGB*, *DDC*, *NCAM1*, *DIO3*, *TH*), suggesting a role in reinforcing NED [26,27,28,29]. Concurrently, INSM1 suppressed genes across four key pathways: cell cycle regulation (*CDC25A*, *RRM2*) [30,31], methionine metabolism (*AHCY*, *MAT2α*) [32,33], transcriptional control (*MYBL2*, *EZH2*) [34,35], and oncogenic signaling (*ALK*, *LINC011667*) [36,37]. Notably, ALK downregulation is significant given its established role in NB pathogenesis and poor prognosis. Additional differentially expressed genes with diverse biological functions include *TRIM29*, *FAM111B*, *CENPU*, and *PCLAF* [38,39,40,41].

## 4. Discussion

We identified the transcription factor, INSM1 as a key regulator of NB malignancy [9]. INSM1 is highly expressed in NB tumors and in developing SA-lineage tissues, particularly during NED of chromaffin cells. In high-risk NB, elevated INSM1 expression reflects a developmental arrest, with tumor cells retaining an immature, progenitor-like phenotype. INSM1 promotes oncogenesis through transcriptional regulation and potential extranuclear functions, activating signaling pathways that favor proliferation and tumor progression over terminal differentiation. We previously demonstrated that N-Myc transcriptionally activates INSM1, while INSM1 stabilizes N-Myc protein, forming a feed-forward oncogenic loop [8]. Co-expression of both genes correlates strongly with poor patient outcomes [12]. Based on these findings, we hypothesize that INSM1 contributes to NB aggressiveness by maintaining a NE progenitor-like state, sustaining poorly differentiated tumor populations, and activating oncogenic transcriptional programs. We investigated the functional role and regulatory mechanisms of INSM1 in NB. To do so, we developed luciferase-based screening platforms driven by the *INSM1*-promoter in two NB cell lines (BE2-M17 and IMR-32) [19]. Using this system, we identified 5′-iodotubercidin (5′-IT), an adenosine kinase (ADK) inhibitor, as a compound that suppresses *INSM1* gene expression and inhibits NB cell growth [11].

Our study identifies INSM1 as a critical regulator of NB malignancy and differentiation status. INSM1, a zinc-finger transcription factor originally discovered in an insulinoma subtraction library, is highly expressed in embryonic NE precursors and NE tumors [42]. In NB, elevated INSM1 expression reflects a developmental arrest, maintaining tumor cells in an immature, progenitor-like state. This is supported by our transcriptomic analysis, where INSM1 overexpression in SH-SY5Y cells induces a NED program, upregulating NE markers (*CHGA*, *CHGB*, *DDC*, *NCAM1*) and thyroid hormone-related genes (*DIO3*, *TH*), while concurrently suppressing cell cycle regulators (*CDC25A*, *RRM2*), methionine metabolism genes (*AHCY*, *MAT2α*), transcriptional control (*MYBL2*, *EZH2*) and oncogenic drivers (*ALK*, *LINC011667*). These findings suggest that INSM1 orchestrates a transcriptional network that promotes NE identity while modulating proliferation and metabolic pathways, reinforcing its role as a master regulator of NB cell immaturity and aggressiveness.

Mechanistically, INSM1 regulation is closely linked to epigenetic and metabolic pathways. Our analysis revealed strong enrichment of H3K4me3 at the *INSM1* locus in INSM1-positive NB cells, indicating an active chromatin state. This mark, along with H3K4me2, suggests involvement of LSD1, which fine-tunes *INSM1*-promoter accessibility through selective demethylation. Furthermore, perturbations in the methionine cycle significantly impact INSM1 expression. Using a luciferase-based *INSM1*-promoter assay, we identified 5′-IT, an ADK inhibitor, as a potent suppressor of INSM1 expression and NB cell growth [11]. ADK is a key enzyme in adenosine metabolism that indirectly influences methylation-dependent epigenetic processes. Its dual role arises from its ability to regulate adenosine levels and thereby control the methionine cycle flux. ADK serves as a metabolic-epigenetic link—by controlling adenosine and SAH levels, it influences methylation reactions and chromatin structure, impacting gene expression programs in NB and other cancers. ADK regulates adenosine clearance, influencing SAH hydrolysis and methylation capacity. Inhibition of ADK by 5′-IT likely disrupts adenosine homeostasis, leading to SAH accumulation and methyltransferase inhibition, thereby altering histone methylation at the *INSM1*-promoter [17]. Conversely, FIDAS-5, a MAT2α inhibitor that reduces SAM synthesis, acting on methylation from an opposite angle as compared to the ADK inhibitor, which paradoxically enhanced *INSM1* promoter activity, highlighting the complexity of methionine cycle flux in epigenetic regulation. Additionally, DZNep, an SAHH inhibitor promotes SAH accumulation and suppresses NB proliferation, consistent with its role in modulating methylation-dependent chromatin remodeling. Potentially, depletion of SAM synthesis favors *INSM1*-promoter activation, whereas SAH accumulation suppresses *INSM1* promoter activity. These findings underscore a metabolic-epigenetic axis in NB, where methionine cycle enzymes and methylation dynamics converge to regulate INSM1 expression and NB tumor phenotype. However, the mechanistic evidence of SAM/SAH index shift during INSM1 expression remains indirect.

INSM1 was identified as a potent NE biomarker highly elevated in NE and neuroepithelial neoplasms [43]. Its temporal expression in developing NE precursors defines the NED state, and dysregulation contributes to the poorly differentiated phenotype characteristic of high-risk NB. Our results position INSM1 as a central node linking developmental arrest, epigenetic regulation, and metabolic signaling in NB. Its overexpression drives transcriptomic reprogramming toward NE identity, enhances cellular invasiveness, and correlates with advanced disease staging. Targeting INSM1 directly or through upstream metabolic and epigenetic pathways offers a promising therapeutic strategy for high-risk NB. Future studies should elucidate the precise molecular interactions between INSM1, chromatin modifiers (e.g., LSD1, EZH2), and methionine cycle enzymes, as well as explore combination therapies that exploit these vulnerabilities.

## 5. Conclusions

NB is a biologically heterogeneous pediatric cancer driven by disrupted differentiation of neural crest-derived sympathetic adrenergic precursors. Our findings suggest INSM1 as a critical regulator of NB malignancy, maintaining an immature NE state and cooperating with N-Myc in a feed-forward oncogenic loop. Using an *INSM1*-promoter luciferase screening platform, we identified 5′-IT, an adenosine kinase inhibitor, as a compound that suppresses INSM1 expression and inhibits NB cell growth, implicating adenosine metabolism and methionine cycle flux in *INSM1* gene regulation. These results suggest that targeting INSM1 and its metabolic regulators may offer novel therapeutic strategies for high-risk NB.

## Figures and Tables

**Figure 1 biology-15-00022-f001:**
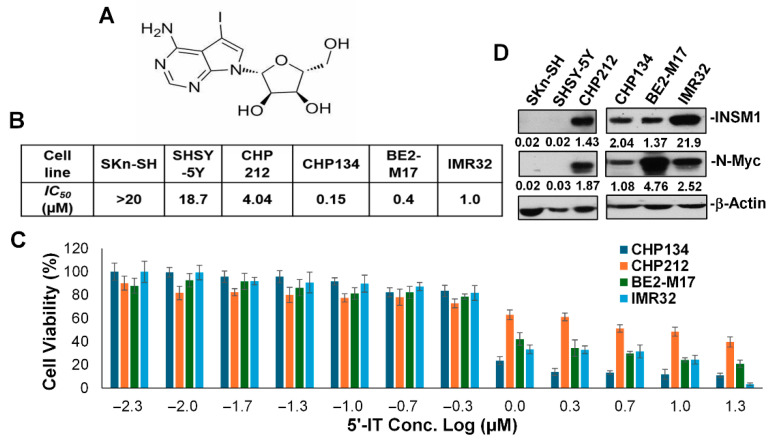
Effects of 5′-IT on INSM1 expression and cell viability in NB cells. (**A**) Chemical structure of 5′-IT. (**B**) Six human NB cell lines were treated with increasing concentrations of 5′-IT (5 nM to 20 μM) for 48 h, and cell viability was assessed using the MTS assay (*n* = 8); IC_50_ values were determined as follows: SKn-SH (>20 μM), SHSY-5Y (18.7 μM), CHP212 (4.04 μM), CHP134 (0.15 μM), BE2-M17 (0.4 μM), and IMR-32 (1.0 μM). (**C**) Dose-dependent cell viability curves for four sensitive NB cell lines (CHP134, CHP212, BE2-M17, IMR-32) show reduced viability with increasing 5′-IT concentration. (**D**) Western blot analysis of INSM1 and N-Myc protein levels (ratios) in all six cell lines were quantified with β-actin used as a loading control.

**Figure 2 biology-15-00022-f002:**
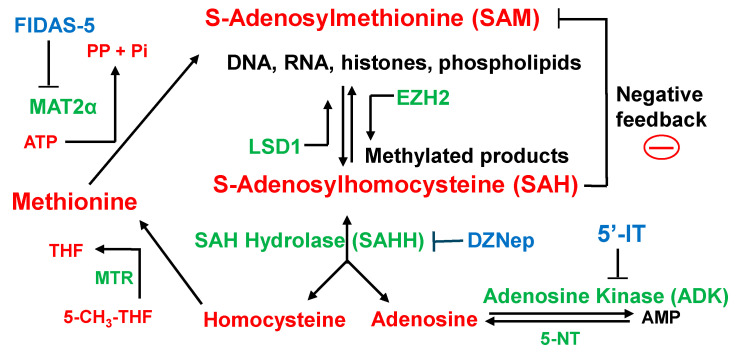
Schematic representation of the methionine cycle pathway highlighting key metabolites, enzymes, and inhibitors. The diagram illustrates the reversible reaction catalyzed by S-adenosylhomocysteine hydrolase (SAHH) and the inhibition of methionine adenosyl-transferase II alpha (MAT2α), a SAM-dependent methyltransferase (shown by flow symbol). Metabolites involved in the pathway shown in red include methionine, S-adenosylmethionine (SAM), S-adenosylhomocysteine (SAH), homocysteine, adenosine, ATP, inorganic phosphate (Pi), pyrophosphate (PP), tetrahydrofolate (THF), and 5-methyl-THF (5-CH3-THF) (flow symbol). Metabolic enzymes shown in green include 5′-nucleotidase (5-NT), adenosine kinase (ADK), SAHH (SAH hydrolase), EZH2 (histone methyltransferase), LSD1 (histone demethylase), MAT2α, and methionine synthase (MTR). Enzyme inhibitors shown in blue (block symbol) include 5′-iodotubercidin (5′-IT, ADK inhibitor), DZNep-HCl (SAHH inhibitor), and FIDAS-5 (MAT2α inhibitor).

**Figure 3 biology-15-00022-f003:**
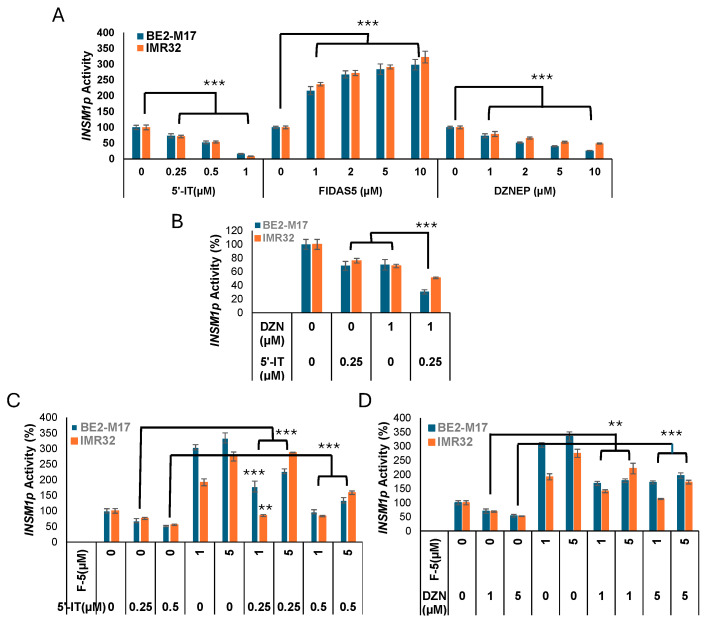
Effects of FIDAS-5, DZNep-HCl, and 5′-IT on *INSM1* promoter activity in IMR32 and BE2-M17 NB cells. (**A**) Cells were treated with increasing concentrations of FIDAS-5 (0–10 µM), DZNep-HCl (0–10 µM), or 5′-IT (0–2 µM) individually for 24 h, and *INSM1* promoter activity was measured. (**B**–**D**) Cells were treated with various combinations of FIDAS-5 (F-5), DZNep-HCl (DZN), and 5′-IT for 24 h to assess combinatorial effects on *INSM1* promoter activity. Eight sample repeats (*n* = 8) plotted against control and indicated as SD (** as *p <* 0.01, *** as *p* < 0.001) using *t*-test.

**Figure 4 biology-15-00022-f004:**
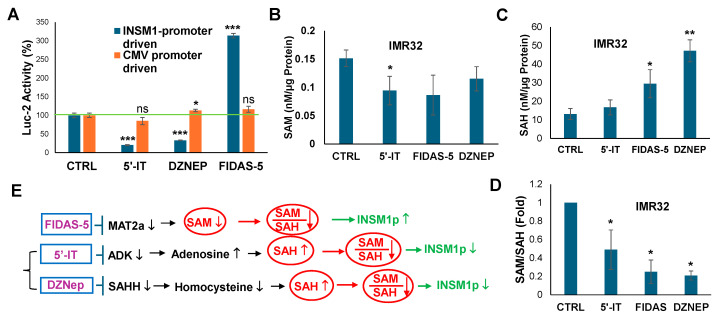
Effects of methionine cycle flux inhibitor on NB cells. (**A**) *INSM1p*-driven-*Luc2* versus *CMVp*-driven-*Luc2* were treated with FIDAS-5, 5′-IT or DZNep. 5′-IT and DZNep inhibit *INSM1* promoter activity, whereas FIDAS-5 enhances INSM1 promoter activity. In contrast to the *CMVp*-driven luciferase assay (*n* = 3). Green line indicates 100% control activity. (**B**–**D**) IMR32 cells treated with 5′-IT, DZNep, or FIDAS-5 for 48 h and subjected to SAM, SAH ELISA assay. Three sample repeats (*n* = 3) plotted against control and indicated as SD (* as *p* < 0.05, ** as *p* < 0.01, *** as *p* < 0.001) using *t*-test. (**E**) Scheme of 5′-IT, DZNep, or FIDAS-5 effect on *INSM1* promoter activity (green arrow indicates promote activity) via disruption of SAM/SAH methylation index.

**Figure 5 biology-15-00022-f005:**
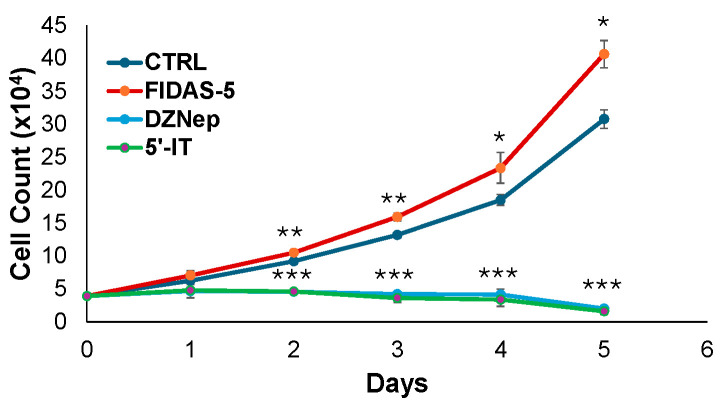
Functional impact of methionine cycle inhibitors on BE2-M17 cell proliferation. Cell growth was assessed using trypan blue exclusion over a 5-day period following treatment with FIDAS-5 (1.0 μM), DZNep (10 μM), or 5′-IT (1.0 μM). Cell counts were recorded daily and plotted against time. Significant differences compared to control are indicated (* as *p* < 0.05, ** as *p* < 0.01, *** as *p* < 0.001).

**Figure 6 biology-15-00022-f006:**
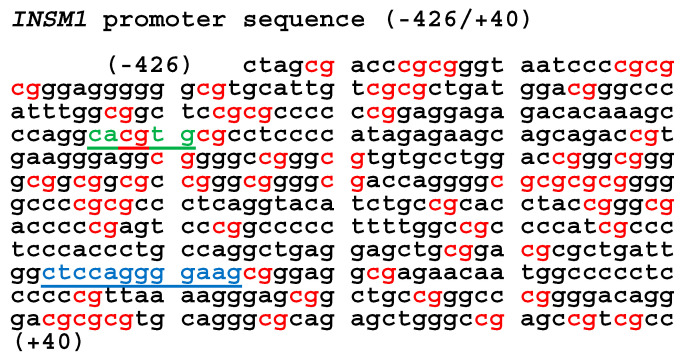
*INSM1*-promoter sequence (−426/+40) and regulatory elements. The *INSM1* promoter, linked to a luciferase reporter and used in transfected BE2-M17 or IMR-32 stable cell lines, contains 56 *CpG* sites (highlighted in red). A canonical *E-box* sequence (*cacgtg*, green underline) marks the N-Myc binding site. DNA methylation within this *CpG*-rich region (red) may play a role in regulating INSM1 expression in NB cells. Additionally, a putative INSM1 autoregulatory binding site (*ctccaggggaag*, blue underline) is located at the proximal promoter region (−78 to −67 bp), contributing to self-suppression of *INSM1* promoter activity.

**Figure 7 biology-15-00022-f007:**
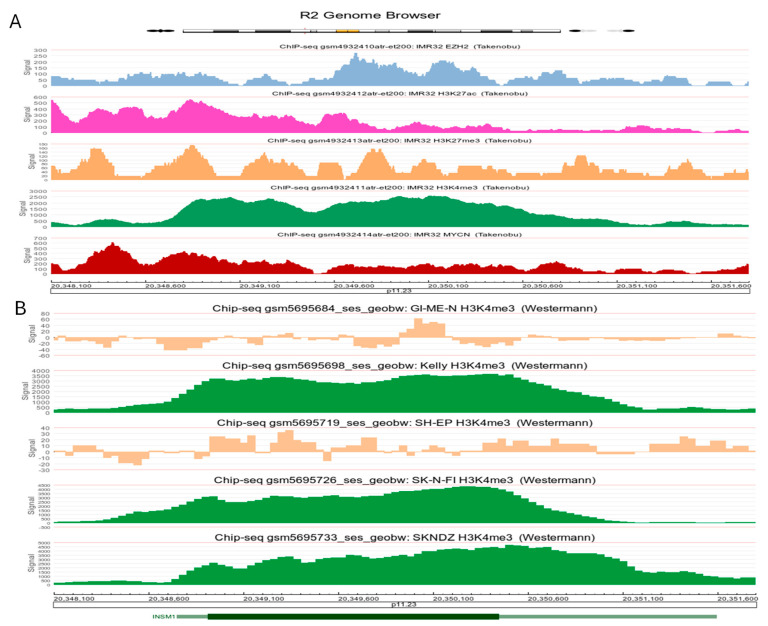
Epigenetic landscape of the *INSM1* gene locus in NB cells. (**A**) *ChIP-seq* datasets (GSE162057) from IMR-32 cells (Takenobu’s group) show enrichment of various epigenetic marks and transcription factors at the *INSM1* gene locus (between 20,348,100 -20,351,600). EZH2, H3K27ac, H3K27me3, H3K4me3, and MYCN were analyzed with R2 Genomics Analysis and Visualization platform. Each color represents different marks enriched in *INSM1* gene locus. (**B**) Comparative analysis by Westermann’s group highlights H3K4me3 profiles (GSE189174) across INSM1-positive (Kelly, SK-N-FI, SKNDZ, green color) and INSM1-negative (GI-ME-N, SH-EP, light brown color) NB cell lines. INSM1-expressing cells exhibit pronounced epigenetic modifications, particularly at promoter and enhancer regions, suggesting a regulatory role in gene activation and/or suppression.

**Figure 8 biology-15-00022-f008:**
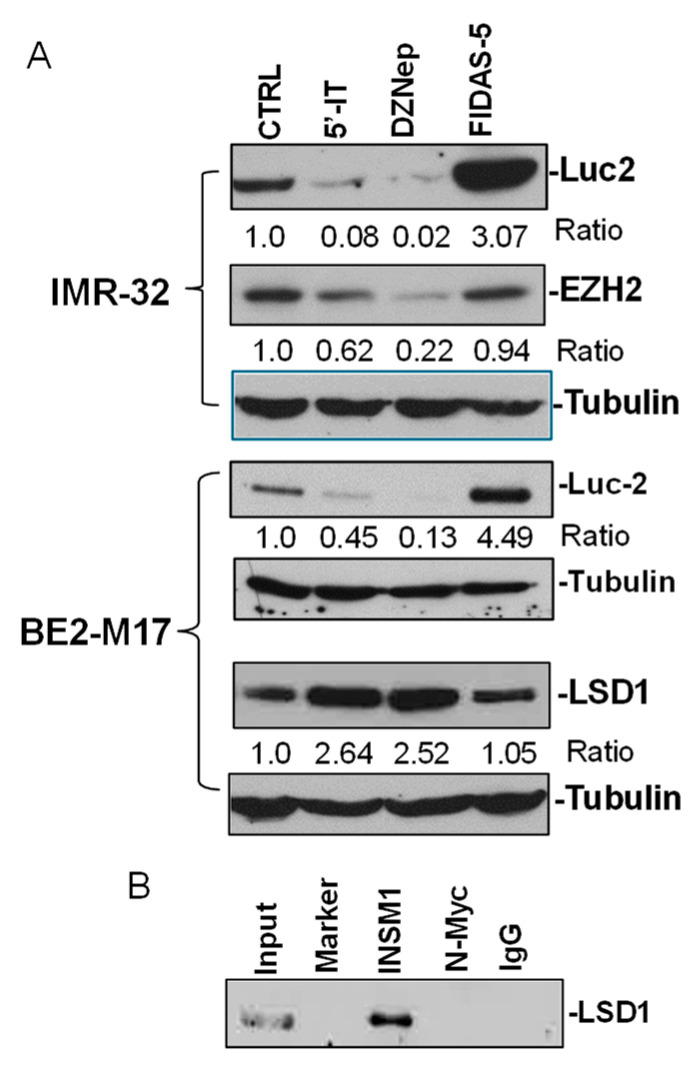
Recruitment of EZH2 and LSD1 to the *INSM1* gene locus. (**A**) Western blot analysis of Luc2, EZH2, and LSD1 protein levels in IMR-32 and BE2-M17 cells treated with 5′-IT, DZNep, or FIDAS-5. Protein expression levels were normalized to tubulin and are shown as relative expression ratios. EZH2 expression was reduced by DZNep and 5′-IT, while LSD1 levels increased under the same treatments. (**B**) Co-immunoprecipitation of INSM1 and LSD1 in BE2-M17 cells. Cell lysates were pre-cleared with agarose protein G-beads and immunoprecipitated using anti-INSM1, anti-N-myc, or control IgG antibodies. The immunoprecipitated complexes were analyzed by SDS-PAGE followed by Western blotting with anti-LSD1 antibody. A 12.5% input lysate served as positive control, and IgG served as negative control.

**Figure 9 biology-15-00022-f009:**
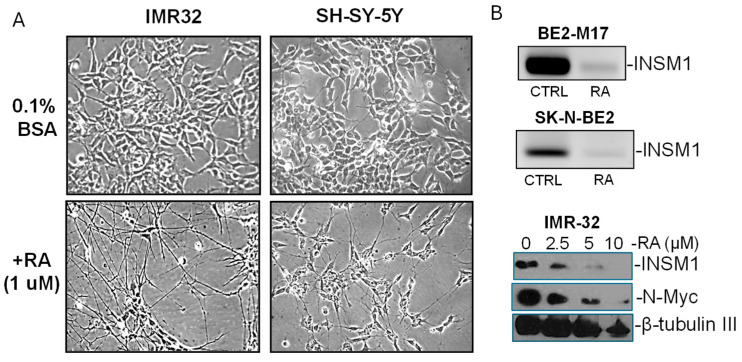
Retinoic acid (RA) promotes neuronal differentiation in NB cell lines. (**A**) Phase-contrast microscopy of IMR-32 and SH-SY-5Y cells cultured in RPMI-1640 medium with 0.1% bovine serum albumin (BSA), either untreated (top row) or treated with 1 µM RA for 5 days (bottom row) (*n* = 3). RA treatment induces prominent neurite outgrowth, indicative of neuronal differentiation. (**B**) Molecular analyses of RA-treated NB cells. Quantitative PCR shows reduced INSM1 mRNA levels in BE2-M17, and SK-N-BE-2 cells compared to untreated controls. Western blot analysis of IMR-32 cells treated with increasing concentrations of RA (0, 2.5, 5, and 10 µM) reveals a dose-dependent decrease in INSM1 and N-Myc protein levels, neuronal marker β-tubulin III serves as a loading control.

**Figure 10 biology-15-00022-f010:**
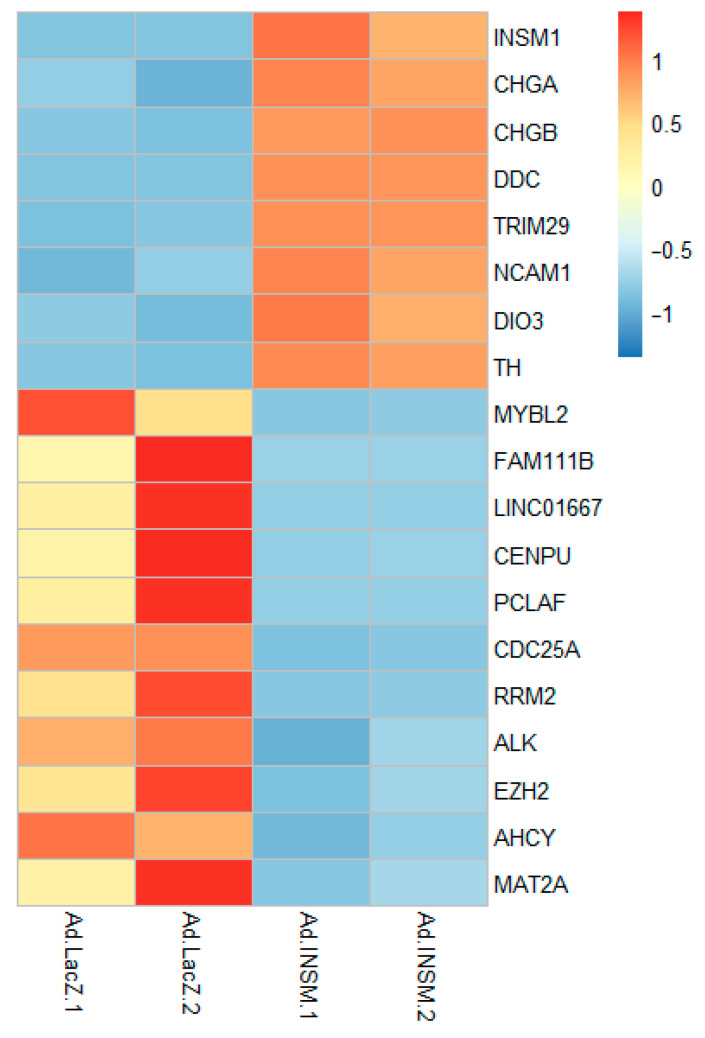
INSM1 overexpression alters transcriptomic profile in SH-SY5Y cells toward NE differentiation. SH-SY5Y cells were infected with *Ad-INSM1* or *Ad-LacZ* control adenovirus for 48 h, followed by transcriptomic analysis (*n* = 2). INSM1-overexpression induced upregulation of six NE and thyroid hormone-related genes: *CHGA*, *CHGB*, *DDC*, *NCAM1*, *DIO3*, and *TH*, indicating a shift toward NE differentiation. Concurrently, INSM1 suppressed expression of genes involved in cell cycle regulation (*CDC25A*, *RRM2*), methionine metabolism (*AHCY*, *MAT2α*), transcriptional control (*MYBL2*, *EZH2*), and oncogenic signaling: *ALK*, *LINC011667*. Additional differentially expressed genes with diverse biological functions include *TRM29*, *FAM1118B*, *CENPU*, and *PCLAF*.

## Data Availability

The data are available from the authors on reasonable request. Data availability and deposition raw and processed *RNAseq* data generated in this study have been deposited in the NCBI Gene Expression Omnibus (GEO) under accession number GSE312485. Raw sequencing reads (FASTQ format) were submitted via GEO’s Sequence Read Archive (SRA) pipeline using FTP/Aspera transfer. Each sample was annotated with metadata including organism, tissue source, library preparation method, sequencing platform, and experimental condition, following GEO’s metadata template requirements.

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
