# Peer review of "Regulation of INSM1 Gene Expression and Neuroendocrine Differentiation in High-Risk Neuroblastoma"

_biology, 2025, doi:10.3390/biology15010022_

Round 1
Reviewer 1 Report
Comments and Suggestions for Authors
Dear Dr Author, thank you for your hard work. I have a suggestion that improves the paper's quality
Introduction:
1) I suggest updating the gene INSM1 implication in cancer’s aggressiveness and poor differentiation
2) 70 Line: extend the description, add a suitable reference
3) Line 70: I suggest that authors enhance their hypothesis; Add a few lines
4) SAHH inhibitor D9 role can be extended in the context of epigenetic regulation.
5) I suggest highlighting the novelty of the study
Methods:
6) The author should clearly state the Cell line selection criteria
Results:
7) Figure 1 - I suggest that authors extend the explanation based on the key findings
8) 276-179 Lines: Require improvement
9) Lines: 308-310 need to be revised/updated
10) 3.5. Functional impact of methionine cycle inhibitors: I suggest the author add more details
11) Lines: 425-430. I suggest to authors that this part become concise
12) 517-519 lines: suggest including more details based on the results
Discusssion
13) Include some lines regarding the INSM1 gene epigenetic mechanism
S-adenosylhomocysteine (SAH), a byproduct of methylation
14) ADK plays a dual role in the context of epigenetic alteration, which may also include (according to current understanding
15) I suggest that authors improve the INSM1 Gene epigenetics aspect, which may include a more concise summary.
16) Specifically, Epigenetic modifications of the INSM1 gene locus must be more informative with arguments.
17) Additionally, INSM1 overexpression linkage with the transcriptomic profile must be mentioned.
18) "INSM1 promotes oncogenesis through transcriptional regulation" becomes more informative; additionally, include the INSM1 expression link to cellular invasiveness and staging
19) I suggest that authors, based on the key findings, update the whole Discussion part (according to their views)
20) include a summary paragraph (even short) regarding the INSM1 Gene Expression alteration as a potential/unique biomarker and therapeutic target What is the perspective episignature associated with INSM1 expression
21) It would be better to include future perspectives
22) The conclusion must be clearly formulated
Author Response
Responses to Reviewer #1
Dear Dr Author, thank you for your hard work. I have a suggestion that improves the paper's quality.
Comments (bold) and responses are listed as follows: Responses are incorporated in the revised manuscript with yellow highlighted.
Introduction:
1) I suggest updating the gene INSM1 implication in cancer’s aggressiveness and poor differentiation:
2) 70 Line: extend the description, add a suitable reference:
3) Line 70: I suggest that authors enhance their hypothesis; Add a few lines:
NB presents a heterogeneous pediatric malignancy characterized by variable differentiation and clinical outcomes, with poorly differentiated tumors often associated with high-risk disease and aggressive behavior. Recent evidence highlights the transcription factor INSM1 as a critical regulator in NE development and tumor biology (9). Our preliminary data demonstrate that INSM1 expression is markedly elevated in poorly differentiated, high-risk NB tumors compared to more differentiated counterparts (12). Functionally, INSM1 appears to sustain the NED state, promote tumor cell proliferation, and enhance malignant potential, thereby contributing to aggressive disease phenotypes (8). These findings suggest that INSM1 not only serves as a biomarker for poor prognosis but may also represent a promising therapeutic target for intervention in aggressive NB subtypes.
4) SAHH inhibitor D9 role can be extended in the context of epigenetic regulation:
Normally, SAHH hydrolyzes SAH into adenosine and homocysteine. This prevents SAH accumulation, which otherwise inhibits methyltransferases. When SAHH is inhibited by D9 (an analog of DZNep), SAH levels rise (15). SAH is a potent feedback inhibitor of SAM-dependent methyltransferases, reducing methylation reactions across DNA, RNA, and proteins (16). This leads to global hypomethylation, affecting gene expression and epigenetic states.
5) I suggest highlighting the novelty of the study:
While INSM1 has been recognized as a marker in NE tumors, its functional role in NB progression remains largely unexplored. Our study is the first to demonstrate that INSM1 is not merely a diagnostic marker but an active driver of aggressive phenotypes at high-risk, poorly differentiated NB. By linking INSM1 expressions to maintenance of the NED state and enhanced tumor proliferation, we uncovered a previously uncharacterized mechanism contributing to NB malignancy. This positions INSM1 as both a prognostic biomarker and a potential therapeutic target, offering new avenues for precision treatment in aggressive NB subtypes.
Methods:
6) The author should clearly state the Cell line selection criteria:
Seven human NB cell lines were selected based upon the expression patterns of INSM1 and N-Myc.
Results:
7) Figure 1 - I suggest that authors extend the explanation based on the key findings:
Figure 1 illustrates the impact of 5’-IT on INSM1 expression and its relationship with NB cell sensitivity. Panel A shows the chemical structure of 5’-IT, a compound hypothesized to modulate NED pathways. Western blot analysis (Figure 1D) confirmed that among the six NB cell lines tested, CHP212, CHP134, BE2-M17, and IMR-32 exhibited strong expressions of both INSM1 and N-Myc, whereas SK-N-SH and SH-SY-5Y lacked detectable levels of these markers. This differential expression pattern represents diverse heterogeneous genetic background as INSM1 and N-Myc are associated with poorly differentiated, high-risk NB phenotypes. Subsequent treatment of these cell lines with increasing concentrations of 5’-IT (ranging from 5 nM to 20 μM for 48 hours) revealed a positive but indirect correlation between cytotoxic sensitivity and baseline INSM1/N-Myc expression (Figure 1B, C). These findings support the hypothesis that INSM1 expression could serve as a biomarker for therapeutic response to 5’-IT and highlight the compound’s potential as a targeted agent against high-risk NB.
8) 176-179 Lines: Require improvement:
To identify genome-wide INSM1-binding sites and associated chromatin modifications, we analyze NB cell lines with high endogenous INSM1 expression (IMR-32, Kelly, SK-N-FI, and SKNDZ) and compared them to INSM1-negative lines (GI-ME-N and SH-EP). Chromatin immunoprecipitation followed by sequencing (ChIP-seq) datasets were analyzed to map INSM1 occupancy and its interplay with key epigenetic regulators, including EZH2, histone marks H3K27ac, H3K27me3, H3K4me3, and the oncogene MYCN.
9) Lines: 308-310 need to be revised/updated:
Revised the whole paragraph.
10) 3.5. Functional impact of methionine cycle inhibitors: I suggest the author add more details:
To evaluate the role of methionine cycle flux in NB cell growth, we performed cell proliferation assays following treatment with three distinct inhibitors: FIDAS-5 (MAT2α inhibitor), 5’-IT (ADK inhibitor), and DZNep (SAHH inhibitor). As shown in Figure 5, these compounds exerted differential effects on NB cell proliferation. FIDAS-5 treatment resulted in a significant increase in cell growth, suggesting that inhibition of MAT2α may relieve a regulatory constraint on INSM1 expression, thereby promoting a proliferative phenotype. In contrast, both 5’-IT and DZNep markedly suppressed cell proliferation consistent with their inhibitory effects on INSM1 promoter activity observed in earlier experiment (Figure 3A). These findings indicate that methionine cycle enzymes exert opposing influences on NB cell biology, likely through epigenetic mechanisms involving methylation-dependent regulation of INSM1 and associated oncogenic pathways. Collectively, these results underscore the importance of methionine metabolism in maintaining NB aggressiveness and highlight potential therapeutic opportunities through targeted disruption of this pathway.
11) Lines: 425-430. I suggest to authors that this part becomes concise:
Analysis revealed strong H3K4me3 enrichment at the INSM1 locus in INSM1-positive NB cells, with minimal modification in INSM1-negative lines (Figure 7B). These findings highlight H3K4me3 as a key epigenetic mark linked to INSM1 expression, suggesting LSD1 involvement through binding to H3K4 di- and tri-methylation (H3K4me2 and H3K4me3) (22).
12) 517-519 lines: suggest including more details based on the results:
These findings collectively indicate that INSM1 functions as a critical regulator of NB cell immaturity by maintaining the NED state and supporting proliferative capacity. RA treatment, which induces differentiation in NB cells, significantly downregulates INSM1 expression, suggesting that suppression of INSM1 is an essential molecular event in the transition from an undifferentiated, aggressive phenotype to a more mature neuronal state. Mechanistically, this downregulation may involve epigenetic remodeling at the INSM1 promoter, including changes in histone modifications and transcription factor occupancy, which shift the chromatin landscape toward a differentiation-permissive state. Thus, INSM1 not only serves as a marker of poor differentiation but also represents a functional barrier to RA-mediated maturation, highlighting its potential as a therapeutic target to enhance differentiation-based strategies in high-risk NB.
Discussion
13) Include some lines regarding the INSM1 gene epigenetic mechanism
S-adenosylhomocysteine (SAH), a byproduct of methylation:
ADK regulates adenosine clearance, influencing SAH hydrolysis and methylation capacity. Inhibition of ADK by 5’-IT likely disrupts adenosine homeostasis, leading to SAH accumulation and methyltransferase inhibition, thereby altering histone methylation at the INSM1-promoter (17). Conversely, FIDAS-5, a MAT2α inhibitor that reduces SAM synthesis, acting on methylation from an opposite angle as compared to the ADK inhibitor, which paradoxically enhanced INSM1 promoter activity, highlighting the complexity of methionine cycle flux in epigenetic regulation. Additionally, DZNep, an SAHH inhibitor, suppressed NB proliferation, consistent with its role in modulating methylation-dependent chromatin remodeling. These findings underscore a metabolic-epigenetic axis in NB, where methionine cycle enzymes and methylation dynamics converge to regulate INSM1 expression and NB tumor phenotype.
14) ADK plays a dual role in the context of epigenetic alteration, which may also include (according to current understanding:
ADK is a key enzyme in adenosine metabolism that indirectly influences methylation-dependent epigenetic processes. Its dual role arises from its ability to regulate adenosine levels and thereby control the methionine cycle flux. ADK serves as a metabolic-epigenetic link—by controlling adenosine and SAH levels, it influences methylation reactions and chromatin structure, impacting gene expression programs in NB and other cancers.
15) I suggest that authors improve the INSM1 Gene epigenetics aspect, which may include a more concise summary.
16) Specifically, Epigenetic modifications of the INSM1 gene locus must be more informative with arguments.
Mechanistically, INSM1 regulation is closely linked to epigenetic and metabolic pathways. Our data reveal strong enrichment of H3K4me3 at the INSM1 locus in INSM1-positive NB cells, indicating an active chromatin state. This mark, along with H3K4me2, suggests involvement of LSD1, which fine-tunes INSM1-promoter accessibility through selective demethylation. Furthermore, perturbations in the methionine cycle significantly impact INSM1 expression. Using a luciferase-based INSM1-promoter assay, we identified 5’-IT, an adenosine kinase (ADK) inhibitor, as a potent suppressor of INSM1 expression and NB cell growth (11).
17) Additionally, INSM1 overexpression linkage with the transcriptomic profile must be mentioned.
18) "INSM1 promotes oncogenesis through transcriptional regulation" becomes more informative; additionally, the INSM1 expression links to cellular invasiveness and staging
Our study identifies INSM1 as a critical regulator of NB malignancy and differentiation status. INSM1, a zinc-finger transcription factor originally discovered in an insulinoma subtraction library, is highly expressed in embryonic NE precursors and NE tumors (41). In NB, elevated INSM1 expression reflects a developmental arrest, maintaining tumor cells in an immature, progenitor-like state. This is supported by our transcriptomic analysis, where INSM1 overexpression in SH-SY5Y cells induced a NED program, upregulating NE markers (CHGA, CHGB, DDC, NCAM1) and thyroid hormone-related genes (DIO3, TH), while concurrently suppressing cell cycle regulators (CDC25A, RRM2), methionine metabolism genes (AHCY, MAT2α), transcriptional control (MYBL2, EZH2) and oncogenic drivers (ALK, LINC011667). These findings suggest that INSM1 orchestrates a transcriptional network that promotes NE identity while modulating proliferation and metabolic pathways, reinforcing its role as a master regulator of NB cell immaturity and aggressiveness.
19) I suggest that authors, based on the key findings, update the whole Discussion part (according to their views)
We updated the whole Discussion part based on the previous comments (see the whole Discussion).
20) include a summary paragraph (even short) regarding the INSM1 Gene Expression alteration as a potential/unique biomarker and therapeutic target. What is the perspective epi-signature associated with INSM1 expression
21) It would be better to include future perspectives
We added a summary and perspectives:
INSM1 was identified as a potent NE biomarker highly elevated in NE and neuroepithelial neoplasms (42). Its temporal expression in developing NE precursors defines the NED state, and dysregulation contributes to the poorly differentiated phenotype characteristic of high-risk NB. Our results position INSM1 as a central node linking developmental arrest, epigenetic regulation, and metabolic signaling in NB. Its overexpression drives transcriptomic reprogramming toward NE identity, enhances cellular invasiveness, and correlates with advanced disease staging. Targeting INSM1 directly or through upstream metabolic and epigenetic pathways offers a promising therapeutic strategy for high-risk NB. Future studies should elucidate the precise molecular interactions between INSM1, chromatin modifiers (e.g., LSD1, EZH2), and methionine cycle enzymes, as well as explore combination therapies that exploit these vulnerabilities.
22) The conclusion must be clearly formulated
We added a conclusion section:
NB is a biologically heterogeneous pediatric cancer driven by disrupted differentiation of neural crest-derived sympathetic adrenergic precursors. Our findings highlight INSM1 as a critical regulator of NB malignancy, maintaining an immature NE state and cooperating with N-Myc in a feed-forward oncogenic loop. Using an INSM1-promoter luciferase screening platform, we identified 5’-IT, an adenosine kinase inhibitor, as a compound that suppresses INSM1 expression and inhibits NB cell growth, implicating adenosine metabolism and methionine cycle flux in INSM1 gene regulation. These results suggest that targeting INSM1 and its metabolic regulators may offer novel therapeutic strategies for high-risk NB.
Reviewer 2 Report
Comments and Suggestions for Authors
Chen et al investigated the mechanism of expression regulation of INSM1, a node in N-Myc driven neuroblastoma, using chemicals that influence SAM to SAH conversion. The proposed connection between SAM/SAH balance and epigenetic control of INSM1 expression and NB cell fate is interesting, but I don’t think it’s fully supported by the data. I believe several things need to be clarified to make the claim in the paper stronger:
- Fig 1: The authors claim that the cytotoxic sensitivity to 5’-IT is positively correlated with INSM1 and N-MYC expression level. This is inconsistent with the data. NMYC-, INSM1- cell lines are different from the double positive cell lines. However, the sensitivity of the double positive cell lines are not predicted by INSM1 and NMYC level. For example, CHP134 has the lowest IC50 but has the lowest INSM1 and NMYC levels.
- Fig 2: for pathways that are reversible, please use equilibrium arrow and not resonance arrow. Additionally, in the text, please say clearly which proteins are 5’-IT (based on the figure it’s ADK) and DZNep inhibiting.
- Fig 3: (1) do the authors observe the same effect in IMR-32-Luc2 cells?
(2) I disagree that DZN and 5’-IT have additive effect – it seems to be very small decrease, and statistical analysis is missing. Regarding rescue of DZN by F-5, it is not concentration dependent (unlike in the case of rescue of 5’-IT by F-5), and the authors might want to explain why. Also the authors might want to consider rearrange the orders of each column to make data interpretation a bit easier (i.e. the groups that need to be compared right next to each other).
- Fig 4: (1) Luciferase assay is performed in BE2-M17, but the SAM/SAH is performed in ELISA. The authors might need to justify that.
(2) I am confused why all three inhibitors lead to a decrease in SAM/SAH ratio but have different effect on INSM1 expression.
(3) Please add error bar to panel (F). Please also report the number of repeats, what the error bar represents (STD or SEM), and statistic test for all columns. Please add lines to indicate the groups compared and the type of test used in the figure legend.
Author Response
Responses to Reviewer #2
Chen et al investigated the mechanism of expression regulation of INSM1, a node in N-Myc driven neuroblastoma, using chemicals that influence SAM to SAH conversion. The proposed connection between SAM/SAH balance and epigenetic control of INSM1 expression and NB cell fate is interesting, but I don’t think it’s fully supported by the data. I believe several things need to be clarified to make the claim in the paper stronger:
Comments (bold) and responses are listed as follows: Responses are incorporated in the revised manuscript with yellow highlighted.
- Fig 1: The authors claim that the cytotoxic sensitivity to 5’-IT is positively correlated with INSM1
and N-MYC expression level. This is inconsistent with the data. NMYC-, INSM1- cell lines are different from the double positive cell lines. However, the sensitivity of the double positive cell lines is not predicted by INSM1 and NMYC level. For example, CHP134 has the lowest IC50 but has the lowest INSM1 and NMYC levels.
We modified the explanation of Fig. 1 finding. Quantitation of INSM1 and N-Myc expression levels revealed that expressions are positive but have an indirect correlation between cytotoxic sensitivity and baseline INSM1/N-Myc expression. Fig. 1D was revised with quantitation of INSM1 and N-Myc expression levels.
- Fig 2: for pathways that are reversible, please use equilibrium arrow and not resonance arrow. Additionally, in the text, please say clearly which proteins are 5’-IT (based on the figure it’s ADK) and DZNep inhibiting.
We changed reversible pathways using equilibrium arrow. Fig. 2 was revised. We added in the text “Methionine pathway inhibitors are 5’-iodotubercidin (5’-IT, ADK inhibitor), DZNep-HCl (SAH hydrolase, SAHH inhibitor), and FIDAS-5 (MAT2α inhibitor).”
- Fig 3: (1) do the authors observe the same effect in IMR-32-Luc2 cells?
Yes, we observed the same effect in IMR-32-Luc2 cells. We incorporated the IMR-32-Luc2 data in Fig. 3.
(2) I disagree that DZN and 5’-IT have additive effect – it seems to be very small decrease, and statistical analysis is missing. Regarding rescue of DZN by F-5, it is not concentration dependent (unlike in the case of rescue of 5’-IT by F-5), and the authors might want to explain why. Also, the authors might want to consider rearrange the orders of each column to make data interpretation a bit easier (i.e. the groups that need to be compared right next to each other).
We rearranged Fig. 3 into four sub-Figs (A, B, C, D) with statistical analysis. DZN and 5’-IT additive effects show significant difference in Fig. 3B. Regarding rescue experiments, both 5’IT (Fig. 3C) and DZN (Fig. 3D) show partial but significant rescue effects. Fig. 3 was revised.
- Fig 4: (1) Luciferase assay is performed in BE2-M17, but the SAM/SAH is performed in ELISA. The authors might need to justify that.
Luciferase assay was performed in both BE2-M17 and IMR-32. We showed the luciferase assay in Fig. 3A. In Fig. 4A, we showed INSMp-driven-luc2 versus CMVp-driven-luc2 in response to methionine pathway inhibitors. We chose IMR-32 cells for ELISA assay due to the availability of ELISA kit.
(2) I am confused why all three inhibitors lead to a decrease in SAM/SAH ratio but have different effect on INSM1 expression.
We added a paragraph in discussion section as follows:
ADK regulates adenosine clearance, influencing SAH hydrolysis and methylation capacity. Inhibition of ADK by 5’-IT likely disrupts adenosine homeostasis, leading to SAH accumulation and methyltransferase inhibition, thereby altering histone methylation at the INSM1-promoter. Conversely, FIDAS-5, a MAT2α inhibitor that reduces SAM synthesis, acting on methylation from an opposite angle as compared to the ADK inhibitor, which paradoxically enhanced INSM1 promoter activity, highlighting the complexity of methionine cycle flux in epigenetic regulation. Additionally, DZNep, an SAHH inhibitor promotes SAH accumulation and suppresses NB proliferation, consistent with its role in modulating methylation-dependent chromatin remodeling. Potentially, depletion of SAM synthesis favors INSM1-promoter activation, whereas SAH accumulation suppresses INSM1 promoter activity. These findings underscore a metabolic-epigenetic axis in NB, where methionine cycle enzymes and methylation dynamics converge to regulate INSM1 expression and NB tumor phenotype.
(3) Please add error bar to panel (F). Please also report the number of repeats, what the error bar represents (STD or SEM), and statistic test for all columns. Please add lines to indicate the groups compared and the type of test used in the figure legend.
We added error bar to panel (F now Fig. 4D), report the number of repeats, SD error bar, and t-test against control in the figure legend. Fig. 4 was revised.
Fig. 8 was revised.
Round 2
Reviewer 2 Report
Comments and Suggestions for Authors
The authors have addressed this reviewer's concern nicely and can be accepted for publication.
Author Response
Dear Editor:
We made changes in response to the reviewers’ comments point by point as follows:
Responses are bold:
- All references are relevant to the content of the manuscript.
- Any revisions are highlighted to the manuscript.
- This is the cover letter.
- We double check the references included in the manuscript.
Comments
Reviewer #1’s comments are addressed. Reviewer #2’s key concerns remain only partially resolved. The manuscript requires clearer, more cautious mechanistic interpretation and improved consistency in data presentation. The following issues must be addressed:
Mechanistic claims
Ensure that conclusions indicate the findings suggest rather than prove the proposed SAM/SAH–INSM1 mechanism.
We modified the conclusion as “Our findings suggest INSM1 as a critical regulator of NB malignancy, maintaining an immature NE state and cooperating with N-Myc in a feed-forward oncogenic loop.”
Clarify SAM/SAH explanation
Provide a brief, clear explanation and acknowledge that the mechanistic evidence remains indirect.
We acknowledged that “However, the mechanistic evidence of SAM/SAH index shift during INSM1 expression remains indirect.”
Address INSM1 autoregulatory motif
State explicitly that this promoter element was not experimentally examined.
INSM1 autoregulatory motif was experimentally examined. We added a reference for that experiment (Nucleic Acids Res. 2002 Feb 15;30(4):1038-45).
Strengthen ChIP-seq interpretation
Clarify that the ChIP-seq analyses rely on public datasets and offer supportive, not definitive, evidence.
We added a statement for ChIP-seq interpretation as “ChIP-seq analyses rely on public datasets and offer supportive, not definitive, evidence.’
Revise RA-differentiation statements
Indicate that INSM1 downregulation is associated with, not required for, RA-induced differentiation.
RA treatment, which induces differentiation of NB cells, significantly downregulates INSM1 expression suggesting that suppression of INSM1 is associated with RA-induced differentiation. Down regulation of INSM1 could be a molecular event in the transition from an undifferentiated, aggressive phenotype to a more mature neuronal state.
Ensure consistent statistical reporting
Include n values, error bars, statistical tests, and significance indicators in all figures.
Done.
Update Figure 1 interpretation
State clearly that INSM1/N-Myc expression correlates indirectly, not predictively, with 5’-IT sensitivity.
Updated.